# Yield Performance of Super Hybrid Rice Grown in Subtropical Environments at a Similar Latitude but Different Altitudes in Southwest China

**DOI:** 10.3390/plants14050660

**Published:** 2025-02-21

**Authors:** Peng Jiang, Dingbing Wang, Lin Zhang, Xingbing Zhou, Mao Liu, Hong Xiong, Xiaoyi Guo, Yongchuan Zhu, Changchun Guo, Fuxian Xu

**Affiliations:** 1Key Laboratory of Southwest Rice Biology and Genetic Breeding, Ministry of Agriculture and Rural Affairs, Rice and Sorghum Research Institute, Sichuan Academy of Agricultural Sciences, Deyang 618000, China; pengjiang@scsaas.cn (P.J.); linzhang@scsaas.cn (L.Z.); xingbingzhou@scsaas.cn (X.Z.); maoliu@scsaas.cn (M.L.); hongxiong@scsaas.cn (H.X.); guoxiaoyi@scsaas.cn (X.G.); yongchuanzhu@scsaas.cn (Y.Z.); changchuns@scsaas.cn (C.G.); 2Crop Ecophysiology and Cultivation Key Laboratory of Sichuan Province, Chengdu 611130, China; 3Guzhang County of Agricultural and Rural Affairs, Xiangxi 416300, China; dingbingwang@scsaas.cn

**Keywords:** grain yield, super hybrid rice, altitude, crop growth rate, radiation use efficiency

## Abstract

Investigating the variation in and key factors influencing the yield of super hybrid rice cultivated at different altitudes but within the same latitude provides valuable insights for further improvements in super hybrid rice grain yields. Field and pot experiments were conducted using four rice varieties at the following two altitudinal locations in Sichuan Province, China: Hanyuan (high, 1000 m) and Luxian (low, 300 m). The results indicated that Hanyuan achieved an average grain yield of 13.89 t ha^−1^ in paddy fields, with yields being from 63.6% to 94.2% higher than those at Luxian in the field experiments and from 10.8% to 68.0% higher in the pot experiments. The grain yield was consistently higher in the soil from Hanyuan compared to that from Luxian at the same sites. In the field experiments, the grain yield was influenced by location (L), plant density (P), and variety (V), but there were no significant interactions between these factors. In the pot experiments, the grain yield was significantly impacted by L, soil (S), and the interaction between L and S. Climatic factors, which varied with the altitude of the planting site, played a crucial role in achieving optimal yields of the super hybrid rice. Hanyuan exhibited more cumulative solar radiation with a longer growth duration and lower temperatures and higher soil fertility compared to Luxian. The higher grain yield observed at Hanyuan was linked to increases in panicle numbers, spikelets per panicle, grain filling, pre- and post-heading biomass production, and the harvest index. The variations in biomass production between Hanyuan and Luxian were largely due to differences in pre- and post-heading crop growth rates (CGRs) and pre-heading radiation use efficiency (RUE), which were influenced by differences in the maximum and minimum temperatures and cumulative solar radiation. This study indicated that the differences in the grain yield of super hybrid rice across various ecological sites are primarily influenced by altitude and soil fertility, and further enhancement of the grain yield can be achieved by concurrently increasing biomass production before and after heading through improvements in pre- and post-heading CGR.

## 1. Introduction

Rice is one of the most crucial cereal crops globally, serving as a primary source of energy and nutrition for over half of the world’s population. China stands as the largest consumer of rice, accounting for about 28% of global rice consumption [1]. As such, sustaining high rice productivity in China is vital for meeting the needs of a growing population and ensuring the nation’s food security [2]. In China, rice yields have seen two significant advancements over four decades, from the 1950s to the 1990s. These gains were largely driven by genetic improvements and an increased harvest index, achieved by reducing plant height through the semi-dwarf gene and exploiting heterosis in hybrid rice production [3]. As a result, rice production increased by 59%, despite a reduction in arable land since the 1980s [4]. By the 2000s, many cultivars with super-high yield potential were developed by integrating heterosis with morphological enhancements [5,6]. The introduction of super hybrid rice cultivars has further boosted rice grain yields by approximately 10% [5,7]. Understanding the yield potential and the factors that determine yields in these cultivars is essential for identifying breeding targets and achieving additional increases in yields.

Rice grain yield is influenced not only by the crop’s genetic traits but also by environmental conditions [8,9]. Some of these super rice cultivars, including both hybrid and inbred varieties, have produced yields exceeding 13.5 t ha^−1^ when grown in high-altitude environments [5,6,8,9,10,11,12,13]. Significant differences in rice yields between low- and high-altitude regions highlight the impact of factors such as higher incident solar radiation, cooler temperatures, increased solar radiation during the post-heading stage [8,9,10,11], improved radiation use efficiency, higher crop growth rates, and lower maintenance respiration on achieving greater yields [13,14]. In general, climatic factors like temperature and solar radiation are simultaneously affected by both altitude and latitude. However, most previous studies have overlooked the influence of latitude on rice yield. Notably, there were significant differences in latitude between the high-altitude sites and the reference low-latitude sites in these studies [8,9,10,11,13]. Additionally, rice yields tend to increase with higher latitudes due to the more favorable temperature conditions, with yield improvements generally being more pronounced at high latitudes compared to low latitudes [15,16]. As a result, it has been challenging to isolate the individual contributions of altitude and latitude to the variations in super-high rice yields based on the findings of these studies.

The yield components (including panicle number, spikelet number, grain filling, and grain weight) are influenced by several factors: the number of tillers formed during the vegetative growth period, the number of panicles induced at the end of the vegetative stage, the number of spikelets developed during the panicle formation period, the number of fertile spikelets set during the booting and flowering stages, and the individual grain weight established during the grain filling period [17,18]. Environmental conditions, such as light and temperature, significantly impact these yield components during their respective phases [18]. Rice production results from the interaction between cultivar performance, environmental conditions and crop management practices. Increasing plant density is often recommended as a strategy to reduce external nitrogen inputs while maintaining high grain yields. Dense planting typically increases the number of tillers per unit area, leading to a higher number of panicles and, consequently, greater grain yield. Recent studies by Xie [19] and Jiang [20,21] have shown that increasing plant density is an effective approach to achieving high grain yields in southern China. However, there is no consensus on which yield components are most responsible for variations in grain yield across different environments with varying altitudes. Different studies have highlighted various components or combinations of components. For instance, Ying [8] and Katsura [10] identified panicles per unit area and spikelets per panicle as key to achieving super-high rice yields. Conversely, Jiang [5], Li [11] and Zhong [13] emphasized the importance of panicles per unit area, grain filling, and grain weight in contributing to super-high yields.

Grain yields are fundamentally determined by total biomass production and the harvest index [22]. While both factors contribute to rice yields, numerous field studies have shown that super-high rice yields are primarily attributable to high total biomass production rather than a high harvest index [8,10,11]. Biomass production can be increased by extending the crop growth duration, enhancing the crop growth rate, or a combination of both [23]. This biomass production is significantly influenced by environmental conditions [10,11,13,14]. In super-high-yield environments, the combination of intense solar radiation and favorable temperatures has been identified as a key factor driving high biomass production in super hybrid rice [9,10,11]. Studies by Ying [8] and Zhong [13] found that super-high-yielding rice cultivars exhibited high biomass production capacity during both the pre-heading and post-heading phases. In contrast, Jiang [9] observed that when a comparison was made between high-yielding and medium-yielding environments, the high biomass production capacity in super hybrid rice was primarily evident during the post-heading phase, with less pronounced biomass accumulation during the pre-heading phase.

These inconsistencies in rice yield and biomass production may stem from regional differences. Climatic conditions vary significantly across regions, and these regional variations inevitably affect the performance of rice cultivars [24,25]. Indeed, there were considerable differences in both latitude and altitude among the super-high-yielding environments studied in earlier research [8,9,10,11,13]. However, these studies did not account for the impact of different latitudes on biomass production and its characteristics when comparing low- and high-altitude environments. Overall, the quantitative impact of environmental conditions on rice yield and biomass productivity in super-high-yielding rice under varying ecological conditions has not been fully explored. To advance our understanding, it is crucial to characterize how different environments with varying altitudes but with the same latitude affect biomass production, yield formation, crop growth rate (CGR), and radiation use efficiency among super hybrid rice genotypes. This could inform strategies to further enhance the yield potential of high-yielding rice cultivars.

In this study, we hypothesized that the variation in and key factors influencing the yield of super hybrid rice could be elucidate through conducting both field and pot experiments at different altitudes with same latitude. To test this hypothesis, both field and pot experiments with four rice hybrids cultivars were conducted at two locations in Sichuan Province, China—Luxian County (low altitude) and Hanyuan County (high altitude)—from 2020 to 2021. Notably, both counties share the same latitude, allowing us to focus on the impact of altitude on rice performance. The objectives of this study were threefold: (1) to identify the key yield attributes responsible for yield differences between the two subtropical environments with different altitudes but the same latitude; (2) to clarify the relationships between environmental factors and rice yields by comparing the effects of four plant densities and four hybrid rice cultivars grown in low- and high-altitude environments at the same latitude; and (3) to provide guidance on selecting high-yielding rice cultivars and determining the optimal plant density for achieving super-high yields in different regions.

## 2. Results

### 2.1. Field Experiments

#### 2.1.1. Growth Duration

The growth duration of the super hybrid rice was observed to be longer in Hanyuan compared to Luxian, with an extension of 3–4 days from transplanting (TR) to heading (HD) and 10–11 days from HD to maturity (MA) (Appendix A). Within each site, the variation in duration from TR to HD and from HD to MA was relatively small across the four hybrid rice varieties. Generally, the rice varieties Luyou727 and Nei6you107 exhibited a longer growth duration compared to Deyou4727 and Nei6you6. On average, the growth duration of the super hybrid rice was slightly longer in 2020 than in 2021.

#### 2.1.2. Climatic Conditions and Soil Properties

The average maximum temperature from TR to HD was either similar or slightly higher in Hanyuan compared to Luxian. However, from HD to MA, the temperature was 5.5–7.5 °C lower in Hanyuan than in Luxian (Appendix A). The average minimum temperatures during TR to HD and HD to MA were 1.5–2.1 °C and 5.2–6.1 °C lower in Hanyuan than in Luxian, respectively. In contrast, the cumulative solar radiation during TR to HD and HD to MA was 196.6–326.6 MJ m^−2^ and 8.7–131.6 MJ m^−2^ higher in Hanyuan than in Luxian, respectively. Within each site, the variations in maximum temperature, minimum temperature, and cumulative solar radiation across the four hybrid rice varieties were relatively small or inconsistent. The organic matter, total nitrogen (N), total potassium (K), available N, and available K in the soil from Hanyuan were 58.0%, 62.5%, 15.5%, 58.1%, and 75.9% higher than in the soil from Luxian, respectively, while there was lower available phosphorus in the soil from Hanyuan than in that from Luxian (Appendix A).

#### 2.1.3. Field Grain Yield

Location (L), variety (V), and plant density (P) all had a significant impact on the grain yield of the super hybrid rice varieties in both years (Table 1), However, the interaction effects between these factors on grain yield were not significant. Hanyuan outperformed Luxian in grain yield, with increases of 63.6–66.2% in 2020 and 88.0–94.2% in 2021. Additionally, grain yield improved with a higher plant density. The plant density D1 yielded 2.5–22.0%, more grain compared to D3 and D4, while the difference between D1 and D2 was relatively small or inconsistent. Among the varieties, Luyou727 had the highest grain yield, while Deyou4727 had the lowest. Luyou727 exceeded the other three hybrid rice varieties by 1.8–12.4% in Hanyuan and by 2.1–7.9% in Luxian. On average, the grain yield of the super hybrid rice was 12.4% higher in 2021 than in 2020.

#### 2.1.4. Field Yield Components

Panicles per m^2^ and spikelets per panicle were 23.0–58.0% and 7.6–34.6% higher, respectively, in Hanyuan compared to Luxian (Table 2). As a result, the sink size (spikelets per m^2^) was 59.6–89.8% greater in Hanyuan than in Luxian. Hanyuan also had a 0.6–12.3% higher grain filling percentage, while Luxian produced similar or slightly higher grain weight. In general, the number of panicles per m^2^ in the super hybrid rice increased with a higher plant density, while spikelets per panicle decreased. The sink size also increased with a higher plant density. Differences in grain filling percentage and grain weight among the four plant densities were relatively small or inconsistent. Among all varieties, Luyou727 had the highest spikelets per panicle and sink size. On average, Luyou727 exceeded the other three hybrid rice varieties in panicles per m^2^, spikelets per panicle, sink size, and grain filling percentage by 2.4%, 11.3%, 14.3%, and 0.4%, respectively, but had a 10.2% lower grain weight. In 2021, spikelets per panicle, sink size, and grain filling percentage were 10.6%, 9.3%, and 1.6%, higher than in 2020, respectively, while panicles per m^2^ and grain weight were 1.3% and 0.1% lower than in 2020, respectively.

#### 2.1.5. Biomass Production

Pre- and post-heading biomass production, total biomass production, and the harvest index were 40.3–62.3%, 70.9–106.0%, 48.9–75.4%, and 6.8–14.8% higher in Hanyuan compared to Luxian (Table 3). Generally, pre- and post-heading biomass production under D1 and D2 were similar to or higher than those under D3 and D4. Consequently, total biomass production under D1 and D2 was 6.6–7.4% higher than under D3 and D4. The differences in the harvest index among the four plant densities were relatively small or inconsistent. There were no consistent differences in pre- and post-heading biomass production, total biomass production, and the harvest index among the four hybrid rice varieties. Luyou727 had a similar or higher pre-heading biomass production, total biomass production, and harvest index compared to the other three hybrid rice varieties. On average, pre- and post-heading biomass production, total biomass production, and the harvest index were 6.0%, 21.0%, 10.6%, and 1.0% higher in 2021 than in 2020.

#### 2.1.6. Crop Growth Rate and Radiation Use Efficiency

The pre-heading crop growth rate (pre-CGR) and post-heading crop growth rate (post-CGR) were 30.5–63.9% and 16.1–78.9% higher in Hanyuan compared to Luxian (Table 4). The pre-CGR and post-CGR of the super hybrid rice under D1 and D2 were similar to or higher than those under D3 and D4. There were no consistent differences in pre-CGR and post-CGR among the four hybrid rice varieties. Pre-heading apparent radiation use efficiency (pre-RUE) and post-heading apparent radiation use efficiency (post-RUE) were 20.3–42.2% and 23.9–107.4% higher in Hanyuan compared to Luxian (Table 4). The differences in pre-RUE and post-RUE across the four plant densities were relatively small or inconsistent. However, pre-RUE and post-RUE under D1 and D2 were higher than those under D3 and D4 by 3.8–8.0% and 9.1–12.6%, respectively. Similarly, the differences in pre-RUE and post-RUE among the four super hybrid rice varieties were relatively small or inconsistent.

#### 2.1.7. Correlations Between Grain Yield and Indexes

The correlations between grain yield (GY) and various factors, including panicles per m^2^ (PP), spikelets per panicle (SPP), spikelets per m^2^ (SP), percentage of grain filling (GF), grain weight (GW), pre-heading biomass (PB1), post-heading biomass (PB2), total biomass (TB), the harvest index (HI), pre-heading crop growth rate (CGR1), post-heading crop growth rate (CGR2), pre-heading apparent radiation use efficiency (RUE1), and post-heading apparent radiation use efficiency (RUE2) were analyzed (Figure 1). Grain yield showed significant positive correlations with yield components (PP, SPP, SP, and GF), biomass (PB1, PB2, and TB), crop growth rate, and radiation use efficiency. Conversely, grain yield was significantly negatively correlated with the mean maximum temperature from heading to maturity (Max.T2), mean minimum temperature from transplanting to heading (Min.T1), and mean minimum temperature from heading to maturity (Min.T2). Additionally, grain yield was significantly related to cumulative solar radiation from transplanting to heading and from heading to maturity. These findings indicate that yield components, biomass production, temperature, and cumulative solar radiation contributed to the large differences in grain yield between the two locations with different altitudes but the same latitude. Yield components, biomass production, CGR, and RUE were significantly negatively correlated with temperature (except for the mean maximum temperature from transplanting to heading) but significantly positively correlated with cumulative solar radiation from transplanting to heading and from heading to maturity. This suggests that favorable environmental conditions are crucial for promoting plant growth, development, and high yield formation in super hybrid rice.

### 2.2. Pot Experiment

#### 2.2.1. Pot Grain Yield

Grain yield was significantly influenced by location (L) and soil (S) but not by variety (V). A significant interaction was observed between L and S, whereas the interactive effects of L × V and S × V on grain yield were not significant (Table 5). However, the interactive effects of L × S × V were significant for grain yield in 2021 but not in 2020. The grain yield of the super hybrid rice was 10.8–68.0% higher in Hanyuan than in Luxian for the soil from Hanyuan and 30.3–38.1% higher for the soil from Luxian. Within the same location, the grain yield was 2.3–33.9% higher for the soil from Hanyuan compared to soil from Luxian in Hanyuan and 3.8–27.5% higher in Luxian. The differences in grain yield among the four super hybrid rice varieties were relatively small or inconsistent. Luyou727 recorded the highest grain yield among the four varieties, except for the soil from Luxian at the Luxian site in 2022.

#### 2.2.2. Pot Yield Components

The plants grown in Hanyuan had 34.7% more panicles per hill, a 2.7% higher grain filling percentage, and a 5.0% greater grain weight compared to those grown in Luxian, although they had 7.9% fewer spikelets per panicle (Table 6). Under the same ecological conditions, the soil from Hanyuan produced 2.8–21.4% more panicles per hill and 12.2–26.4% more spikelets per panicle than the soil from Luxian. The differences in grain filling percentage and grain weight between the two soils were relatively small. Differences in panicles per hill, spikelets per panicle, and grain filling percentage among the four hybrid rice varieties were either inconsistent or relatively minor. Luyou727 generally had the highest number of panicles per hill and spikelets per panicle but the lowest grain weight among the four hybrid rice varieties.

#### 2.2.3. Biomass Production and Harvest Index

On average, biomass production at the maturity stage was 9.0–41.6% higher in Hanyuan compared to Luxian, while the difference in the harvest index between the two locations was relatively small (Table 6). Under the same ecological conditions, biomass production and the harvest index in the soil from Hanyuan were 0.4–29.0% and 1.8–6.1% higher, respectively, than in the soil from Luxian at the Hanyuan location. At the Luxian location, the plants grown in the soil from Hanyuan accumulated 4.0–36.3% more biomass than those plants grown in the soil from Luxian, although the harvest index was 0.2–3.5% lower in the soil from Hanyuan compared to the soil from Luxian. The differences in biomass production and harvest index among the hybrid rice varieties were inconsistent or relatively small. The hybrid rice produced 3.1% higher biomass in 2020 than in 2021, but the harvest index was 8.0% lower in 2020 than in 2021.

## 3. Discussion

Previous studies have consistently reported that crops planted at higher altitudes produce significantly higher grain yields compared to those planted at lower altitudes [8,9,10,11]. However, many of these studies also involved substantial differences in latitude between high- and low-altitude sites, which may have confounded the results. The grain yield of hybrid rice has been shown to vary with latitude [15,16], suggesting that previous studies may not have fully accounted for the effects of altitude on rice yield. In the present study, we compared super hybrid rice in two subtropical environments at the same latitude but largely differing in altitude. At the high-altitude site (1000 m above sea level, asl) in Hanyuan, grain yields exceeded 16.69 t ha^−1^ with an average grain yield of 13.89 t ha^−1^, representing a 63.6–94.2% increase compared to the low-altitude site (300 m asl) in Luxian. The highest grain yield is higher than those reported in earlier studies at high-altitude sites [8,9,10,13]. These findings support the notion that large differences in grain yields of super hybrid rice at various ecological sites are primarily related to altitude. Hanyuan, being a high-altitude site, represents a typical super-high-yield environment for rice production. At Hanyuan, the higher altitude provides superior solar radiation and temperature conditions during the rice-growing season (Appendix A). Because the latitude difference between Hanyuan and Luxian is negligible, the variations in solar radiation and temperatures primarily stemmed from altitude differences. The cumulative solar radiation during pre-heading and post-heading was 196.6–326.6 MJ m^−2^ and 8.7–131.6 MJ m^−2^ higher in Hanyuan than in Luxian, respectively, due to the longer growth duration at 3–4 and 10–13 days, respectively; the average maximum and minimum temperatures during post-heading were 5.5–7.5 °C and 5.2–6.1 °C lower in Hanyuan than in Luxian, respectively (Appendix A). The higher solar radiation and lower temperature can easily enhance the grain yield of hybrid rice grown under high-altitude environment [13]. In this study, the correlation analysis revealed that the grain yield was significantly positively correlated with cumulative solar radiation during both the pre- and post-heading periods but significantly negatively correlated with the maximum temperature during the post-heading period and the minimum temperatures during both the pre- and post-heading periods (Figure 1). In the pot experiment, both location (L) and soil (S) significantly affected grain yields, with a notable interaction between the two. The grain yield was 10.8–68.0% higher in Hanyuan compared to Luxian. Similarly, within the same ecological site, the soil from Hanyuan also produced higher grain yields compared to the soil from Luxian. Hanyuan had higher organic matter, total nitrogen, total phosphorus, total potassium, available nitrogen, and available potassium compared to Luxian (Appendix A). This suggests that the higher soil fertility in Hanyuan contributed to the observed differences in grain yield. Considering the results from both the paddy field and pot experiments, it is evident that the differences in grain yield of the super hybrid rice at various ecological sites were primary related to altitude and soil fertility. This finding partially contradicts the results reported by Jiang [5], who attributed the significant difference in grain yield between two subtropical environments primarily to climatic conditions rather than soil fertility. One reason for this discrepancy may be the variation in yield levels between the high-yielding sites studied. In the present study, the rice crops achieved an average grain yield of 13.89 t ha^−1^ at the high-yielding site, whereas Jiang [5] reported an average yield of only 10 t ha^−1^ at their high-yielding site. Previous research has shown that both indigenous soil nitrogen and applied nitrogen are crucial for increasing rice yields from medium (6.5–7.5 t ha^−1^) to high levels (9.0 t ha^−1^). However, for achieving super-high yields (12.0 t ha^−1^), the contribution of indigenous soil nitrogen becomes more significant compared to fertilizer nitrogen [26]. Total nitrogen and available nitrogen in the soil from Hanyuan were 62.5% and 58.1% higher than in the soil from Luxian, respectively (Appendix A). These findings suggest that altitude and soil fertility can account for the yield gap of mid-season rice between high- and low-altitude sites, without the confounding influence of latitude.

The yield advantage observed at Hanyuan compared to Luxian can be attributed to a higher number of panicles per m^2^, an increased number of spikelets per panicle, and a greater percentage of grain filling. Typically, in rice crops, a compensatory compensation mechanism exists between the panicle number per m^2^ and spikelet number per panicle [8]. However, in this study, the higher panicle number per m^2^ at Hanyuan did not result in a lower spikelet number per panicle compared to Luxian. The number of panicles is closely related to the number of tillers, which is regulated by carbohydrate supply [27], while spikelets per panicle are linked to biomass production before heading [8]. Previous research has demonstrated that the compensatory relationship between the panicle number per m^2^ and spikelet number per panicle can be decoupled by increasing biomass accumulation before heading [28]. In our study, biomass production before heading was 40.3–62.3% higher in Hanyuan than in Luxian. Significant positive correlations were found between pre-heading biomass accumulation and both the panicle number and spikelets per panicle (Figure 1). This suggests that increased biomass accumulation before heading was a major factor contributing to the higher panicle number and greater number of spikelets per panicle in Hanyuan. However, the impact of climatic factors during critical growth periods on the yield determination process remains less understood, especially in super-high-yield sites. In this study, cumulative solar radiation before heading was higher in Hanyuan than in Luxian, and the mean minimum temperature before heading was lower in Hanyuan (Appendix A). Lower nighttime temperatures reduce rice crop respiration, while higher daytime solar radiation enhances photosynthesis [10], resulting in a lower respiration-to-photosynthesis ratio and increased biomass accumulation. These findings suggest that the higher panicle number and greater number of spikelets per panicle in Hanyuan can be explained by the combination of higher cumulative solar radiation and lower minimum temperatures before heading. Since we identified the panicle number and spikelets per panicle as key yield-determining factors in Hanyuan, we further analyzed their relationships with climatic variables before heading. A negative relationship was observed between the panicle number and the mean minimum temperature before heading, while a positive relationship was found between the panicle number and cumulative solar radiation before heading (Figure 1). The panicle number showed a stronger correlation with cumulative solar radiation before heading than with the mean minimum temperature before heading, with approximately 95% of the variation in the panicle number explained by cumulative solar radiation and 76% by the mean minimum temperature. These results suggest that cumulative solar radiation before heading is likely the primary environmental determining factor determining the panicle number. Furthermore, the difference in growth duration from transplanting to heading between Hanyuan and Luxian was minimal, reinforcing that daily solar radiation intensity before heading is probably the key environmental determinant for the panicle number.

A cool night temperature alone may increase the number of spikelets per panicle, as elevated nighttime temperatures are known to reduce yield and biomass due to increased respiration loss and premature leaf senescence [29]. Specifically, biomass production decreases by 10% for every 1 °C rise in minimum temperature. In this study, Hanyuan had a mean minimum temperature before heading that was 1.5–2.1 °C lower than Luxian, which resulted in 40.3–62.3% higher biomass production before heading at Hanyuan compared to Luxian. There was a negative relationship between spikelets per panicle and the mean minimum temperature before heading and a positive relationship between spikelets per panicle and cumulative solar radiation before heading (Figure 1). Spikelets per panicle correlated more closely with the mean minimum temperature before heading than with cumulative solar radiation before heading. In contrast, there was a strong negative relationship between spikelets per m^2^ and the mean minimum temperature before heading and a strong positive relationship between spikelets per m^2^ and cumulative solar radiation before heading (Figure 1). These results indicate that the combination of lower mean minimum temperatures and higher cumulative solar radiation before heading likely reduces the respiration-to-photosynthesis ratio, enhancing biomass production. This effect appears to decouple the usual compensatory relationship between the panicle number per m^2^ and spikelet number per panicle.

A significant difference in the percentage of grain filling was observed between Hanyuan and Luxian, regardless of whether the measurements were taken in the field or pot experiments (Table 2 and Table 6). High grain filling percentages can be achieved either by increasing biomass production (source capacity) or by enhancing the transport of assimilates to the grains under favorable environmental conditions [30]. In this study, biomass production pre- and post-heading, total biomass production, and the harvest index (excluding the soil from Luxian in 2020) were higher at Hanyuan compared to Luxian (Table 3 and Table 6). Correlation analysis revealed that the percentage of grain filling was significantly positively correlated with pre- and post-heading biomass production, total biomass production, and the harvest index. Notably, the percentage of grain filling showed a closer correlation with pre-heading biomass production and the harvest index than with post-heading biomass production and total biomass (Figure 1). These results suggest that the higher percentage of grain filling at Hanyuan compared to Luxian was primarily due to a greater source production capacity before heading and more efficient transport of assimilates to the grains.

Higher total biomass production can be achieved by enhancing both pre-heading and post-heading biomass production. Generally, high post-anthesis biomass production and efficient translocation of non-structural carbohydrates are crucial for achieving super-high-yielding rice [31,32]. In this study, the increased total biomass production at Hanyuan compared to Luxian was primarily due to both higher pre-heading and post-heading biomass production. This finding contrasts with previous studies [8,9,10], which reported that super-high-yielding sites typically produced more post-heading biomass than sites with average rice yields. Grain yield showed a significant positive correlation with both pre-heading and post-heading biomass production, as well as total biomass. Notably, grain yield was more closely related to pre-heading biomass and total biomass production than to post-heading biomass production. These results suggest that further improvements in rice grain yield in subtropical environments can be achieved by increasing pre-heading biomass production, aligning with the findings reported by Jiang [5].

Biomass production in rice is influenced by both the growth duration and crop growth rate. Climatic factors affecting these variables, such as solar radiation intensity and temperatures, can differ based on the latitude and altitude of the planting site. In this study, the negligible difference in latitude between Hanyuan and Luxian means that variations in solar radiation intensity and temperatures were primarily due to differences in altitude. The higher pre-heading crop growth rate (pre-CGR) at Hanyuan was the key factor behind its greater pre-heading biomass production compared to Luxian, as the pre-heading growth duration between the two sites was negligible. Meanwhile, the higher post-heading biomass production at Hanyuan was attributed to both a longer post-heading growth duration and a higher post-heading crop growth rate (Table 4 and Appendix A). The higher pre-CGR at Hanyuan was linked to increased canopy photosynthesis and pre-heading radiation use efficiency (pre-RUE), driven by higher cumulative solar radiation and a lower mean minimum temperature. Temperature plays a crucial role in determining crop growth duration, with lower temperatures typically leading to longer growth durations. This explains why the post-heading growth duration was longer at Hanyuan than at Luxian. These findings suggest that the higher pre- and post-heading biomass and total biomass production at Hanyuan compared to Luxian were primarily due to the higher pre- and post-CGR.

## 4. Materials and Methods

### 4.1. Study Sites, Materials and Design

Field experiments were carried out from 2020 to 2021 in the following two locations in Sichuan Province, China: Hanyuan County (29.5° N, 102.6° E, 1000 m above sea level) and Luxian County (29.5° N, 105.4° E, 300 m above sea level). Both locations are at the same latitude but differ in altitude, providing an opportunity to study the impact of altitude on rice cultivation. The soil characteristics of the upper 20 cm surface layer are shown in Appendix A. Soil testing was performed on samples taken from the upper 20 cm layer before transplanting the rice seedlings in 2020, and the soil properties were determined by the method of Lu [33].

Each year, three super hybrid rice varieties—‘Deyou4727’, ‘Luyou727’, and‘Nei6you9’—along with one high-yielding hybrid variety—‘Nei6you107′—were planted at both sites under four different plant densities: D1 (27 hills m^−2^), D2 (23 hills m^−2^), D3 (18 hills m^−2^), and D4 (14 hills m^−2^). The information about the four varieties is shown in Appendix A. The experiment was designed as a split-plot layout, with plant density as the main plot factor and the hybrid rice varieties as the sub-plot factor. Each treatment was replicated three times, with a sub-plot size of 40 m^2^. Pre-germinated seeds were sown in seedbeds on March 26 in Hanyuan County and March 5 in Luxian County during the 2020–2021 growing seasons. The seedlings were transplanted at an age of 30–32 days in Hanyuan County and 33–36 days in Luxian County with two seedlings per hill. Fertilization was standardized across both locations, using urea for nitrogen (N), single superphosphate for phosphorus (P), and potassium chloride for potassium (K) at application rates of 180 kg N ha^−1^, 90.0 kg P_2_O_5_ ha^−1^, and 180 kg K_2_O ha^−1^, respectively. Nitrogen was applied in three splits: 50% as a basal application, 20% at early tillering (7 days after transplanting), and 30% at panicle initiation. Phosphorus was applied entirely as a basal application, while potassium was split 50% as a basal application and 50% at panicle initiation. Water management followed a sequence of flooding, mid-season drainage, re-flooding, and moist intermittent irrigation. Throughout the growing season, intensive control measures were employed to manage insects, diseases, and weeds, ensuring no yield losses occurred due to these factors.

Six hills were sampled from each plot at the full heading (HD) stage and again at maturity in each year of the study. At the HD stage, plant samples were separated into stems, leaves, and panicles. At maturity, after counting the number of panicles, the plant samples were hand-threshed to separate the filled grains from the unfilled spikelets. The filled grains were further separated by submerging them in tap water. Three subsamples of 30 g of filled grains were taken along with all unfilled spikelets for the spikelet count. The dry weight of each plant organ was determined by oven-drying the samples at 70 °C until a constant weight was achieved. Key metrics such as pre-heading, post-heading, and total biomass production, the harvest index, and crop growth rate (CGR) during the pre-heading and post-heading periods were calculated. Additionally, panicles per m^2^, spikelets per panicle, grain filling, and grain weight were measured. Apparent radiation use efficiency (RUE) during the pre- and post-heading periods was calculated by dividing the biomass produced during these periods by the cumulative solar radiation received during the corresponding timeframe. Grain yield was determined from a 5 m^2^ area in the middle of each plot, and the yield was adjusted to a standard moisture content of 13.5%.

### 4.2. Pot Experiments

Pot experiments were also conducted at both Hanyuan County (29.5° N, 102.6° E, 1000 m asl) and Luxian County (29.5° N, 105.4° E, 300 m asl) during the 2020 and 2021 growing seasons in the same paddy fields where the field experiments were carried out in Sichuan Province in southwest China. The soil used in these pot experiments was collected from the top 20 cm layer of the same paddy fields at each location.

Before filling the pots, the soil was air-dried, pulverized, and thoroughly mixed. Each pot was filled with 10 kg of this air-dried soil. One day before transplanting, the pots were filled with tap water, and fertilizers were applied according to the same rates used in the field experiments. The fertilizers were mixed well with the soil in each pot. The same rice cultivars used in the field experiments were planted in the pots, with each cultivar replicated three times and with two pots per replication. The pots were arranged in a completely randomized design, with four hills per pot and two seedlings per hill. To prevent shading, the pots were spaced 25 cm apart. In total, 48 pots were set up at each site, with 24 pots containing soil from the Hanyuan County paddy field and another 24 pots containing soil from the Luxian County paddy field. The pots were kept flooded throughout the entire growing season. Intensive control measures were employed against insects, diseases, and weeds to ensure no yield loss occurred due to these factors.

At maturity, the plants from the pot experiments were sampled and the panicles were counted before being hand-threshed. To distinguish filled spikelets from unfilled ones, the spikelets were submerged in tap water. Three subsamples, each consisting of 30 g of filled grains, along with all unfilled spikelets, were taken to determine the total number of spikelets. The dry weight of each plant organ was measured after the samples were oven-dried at 70 °C until they reached a constant weight. Key metrics such as the number of panicles per hill, spikelets per panicle, grain filling, grain weight, grain yield per hill, biomass production, and the harvest index were then calculated.

### 4.3. Statistical Analysis

Climatic data for the study were obtained from local meteorological bureaus. Statistical analyses were conducted using the Statistix 8 software package (Analytical Software, Tallahassee, FL, USA) with an analysis of variance (ANOVA) approach. For the field experiments, the ANOVA model included the following factors: replication, location (L), plant density (P), variety (V), and interactions between these factors, specifically L × P, L × V, and P×V and the three-factor interaction of L × P × V. For the pot experiments, the ANOVA model similarly included replication, location (L), soil (S), and variety (V), as well as the two-factor interactions L × S, L × V, and S × V and the three-factor interaction L × S × V. The criterion for statistical significance was established at the 0.05 probability level.

## 5. Conclusions

In the field experiments conducted at Hanyuan, the grain yield of the super hybrid rice reached over 16.69 t ha^−1^, with an average yield of 13.89 t ha^−1^. This yield was from 63.6% to 94.2% higher than that achieved at Luxian. Similarly, in the pot experiments, Hanyuan outperformed Luxian, with grain yields that were from 10.8% to 68.0% higher. Additionally, the grain yield was higher in the soil from Hanyuan than in the soil from Luxian when tested within the same ecological site. The grain yield was significantly negatively correlated with the mean maximum temperature from heading to maturity, mean minimum temperature from transplanting to heading, and mean minimum temperature from heading to maturity, but it was significantly related to cumulative solar radiation from transplanting to heading and from heading to maturity. This finding indicated that favorable environmental conditions and higher soil fertility are crucial for promoting plant growth, development, and high yield formation in super hybrid rice. Hanyuan had higher soil fertility and more favorable climatic conditions than Luxian. Our results suggest that differences in grain yields between ecological sites are primarily related to the altitude and soil fertility of the planting location. The superior grain yield at Hanyuan compared to Luxian was attributed to several factors: higher panicle numbers, more spikelets per panicle, a greater percentage of grain filling, increased pre- and post-heading biomass production, and a higher harvest index. The combination of lower mean minimum temperatures and higher cumulative solar radiation before heading at Hanyuan contributed to enhanced biomass production, which decoupled the typical compensatory relationship between the panicle number per m^2^ and spikelet number per panicle. The differences in biomass production between Hanyuan and Luxian were mainly due to variations in pre- and post-heading crop growth rates (CGRs) and pre-heading radiation use efficiency (RUE), driven by differences in maximum and minimum temperatures and cumulative solar radiation. This study suggests that further improvements in rice grain yields can be achieved by simultaneously increasing pre- and post-heading biomass production through enhancing pre- and post-CGR.

## Figures and Tables

**Figure 1 plants-14-00660-f001:**
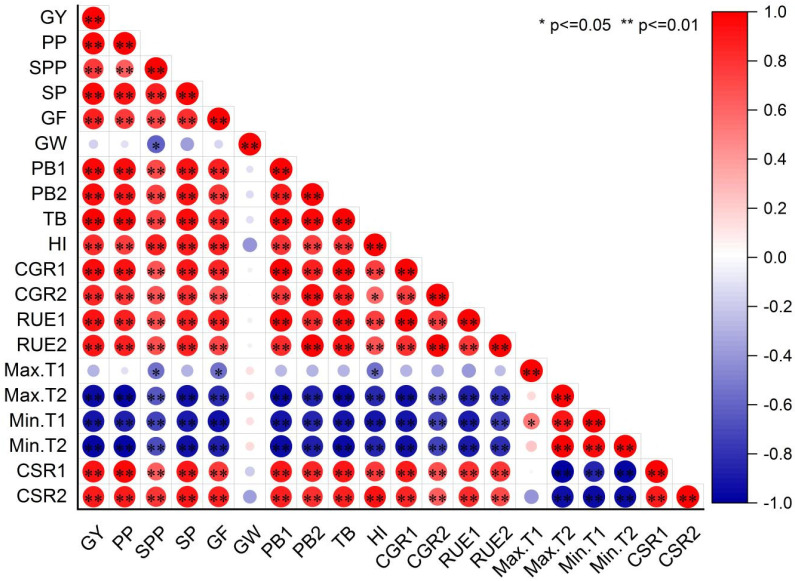
Correlation analysis between indexes and grain yield of four hybrid rice varieties grown at two locations from 2020 to 2021. Data are averaged across four plant densities. Note: GY, grain yield; PP, panicles per m^2^; SPP, spikelets per panicle; SP, spikelets per m^2^; GF, grain filling; GW, grain weight; PB1, pre-heading biomass; PB2, post-heading biomass; TB, total biomass; HI, harvest index; CGR1, pre-heading crop growth rate; CGR2, post-heading crop growth rate; RUE1, pre-heading apparent radiation use efficiency; RUE2, post-heading apparent radiation use efficiency; Max.T1, mean maximum temperature from transplanting to heading; Max.T2, mean maximum temperature from heading to maturity; Min.T1, mean minimum temperature from transplanting to heading; Min.T2, minimum temperature from heading to maturity; CSR1, cumulative solar radiation from transplanting to heading; CSR2, cumulative solar radiation from heading to maturity.

**Table 1 plants-14-00660-t001:** Grain yield (t ha^−1^) of super hybrid rice varieties grown under different plant density in Hanyuan County and Luxian County of Sichuan Province in a field experiment from 2020 to 2021.

Variety	Plant Density (Hills m^−2^)	2020		2021	
		Hanyuan	Luxian	Hanyuan	Luxian
Deyou4727	27	12.53 a	7.88 a	14.88 a	7.94 a
	23	12.18 a	7.74 a	14.42 a	7.68 b
	18	12.22 a	7.25 ab	13.35 ab	7.26 c
	14	12.11 a	6.95 b	12.69 b	6.55 d
	Mean	12.26	7.46	13.84	7.36
Luyou727	27	13.59 b	8.71 a	16.69 a	8.73 a
	23	14.41 a	8.33 ab	16.49 a	8.53 a
	18	13.09 bc	7.79 bc	14.84 b	7.97 b
	14	12.41 c	7.38 c	14.21 b	7.47 c
	Mean	13.38	8.05	15.56	8.18
Nei6you9	27	13.22 a	8.13 a	16.31 a	8.61 a
	23	13.35 a	8.11 a	15.75 a	8.18 b
	18	12.80 ab	7.74 ab	15.44 a	7.64 c
	14	11.87 b	7.33 b	13.64 b	7.06 d
	Mean	12.81	7.83	15.28	7.87
Nei6you107	27	13.47 a	8.25 a	16.16 a	8.62 a
	23	13.21 ab	7.92 ab	15.65 ab	8.28 ab
	18	12.64 bc	7.75 ab	15.10 ab	7.91 bc
	14	12.01 c	7.27 b	13.67 b	7.23 c
	Mean	12.83	7.80	15.14	8.01
Analysis of variance (ANOVA)					
Location (L)		**		**	
Plant density (P)		**		**	
Variety (V)		**		**	
L × P		ns		ns	
L × V		ns		ns	
P × V		ns		ns	
L × P × V		ns		ns	

Note: Within the column for each cultivar, means of plant densities followed by different letters were significantly different according to LSD at *p* = 0.05; ** significant at the 0.01 level based on analysis of variance. ns denotes non-significance based on ANOVA.

**Table 2 plants-14-00660-t002:** Yield components of super hybrid rice varieties grown under different plant densities in Hanyuan County and Luxian County of Sichuan Province in a field experiment from 2020 to 2021.

Year	Variety	Plant Density (Hills m^−2^)	Panicles per m^2^		Spikelets per Panicle		Spikelets per m^2^		Grain Filling (%)		Grain Weight (mg)	
			Hanyuan	Luxian	Hanyuan	Luxian	Hanyuan	Luxian	Hanyuan	Luxian	Hanyuan	Luxian
2020	Deyou4727	27	304.4 a	216.4 a	144.0 b	130.1 bc	43.7 a	28.1 a	93.4 a	86.2 a	29.7 a	30.8 a
		23	241.0 b	211.7 a	171.1 a	120.8 c	41.2 ab	25.6 b	94.6 a	84.6 a	29.6 a	30.9 a
		18	221.0 bc	184.4 b	181.6 a	134.5 b	40.1 ab	24.8 b	93.4 a	86.2 a	29.3 a	30.9 a
		14	212.0 c	160.1 c	185.3 a	150.8 a	39.3 b	24.1 b	94.8 a	85.6 a	29.3 a	30.5 a
		Mean	244.6	193.2	170.5	134.1	41.1	25.7	94.0	85.6	29.5	30.8
	Luyou727	27	268.3 a	219.5 a	196.3 a	142.4 b	52.5 a	31.2 b	91.4 a	84.1 a	26.2 a	27.4 a
		23	265.9 ab	206.6 ab	187.8 a	160.0 a	49.9 a	33.0 a	93.6 a	86.4 a	27.8 a	26.8 b
		18	269.1 a	192.4 b	183.6 a	155.5 ab	49.4 a	29.9 bc	93.5 a	85.8 a	26.1 a	27.3 a
		14	250.3 b	172.9 c	176.9 a	167.0 a	44.2 b	28.9 c	90.9 a	88.0 a	26.5 a	27.4 a
		Mean	263.4	197.9	186.1	156.2	49.0	30.7	92.3	86.1	26.7	27.2
	Nei6you9	27	239 bc	248.0 a	186.5 a	119.0 c	44.6 ab	29.5 a	92.0 ab	83.3 a	29.6 a	30.4 a
		23	269.7 ab	214.3 b	173.3 b	128.1 b	46.8 a	27.4 b	91.9 ab	82.5 a	29.4 a	30.1 a
		18	288.6 a	197.4 c	159.0 c	131.8 b	45.7 ab	26.0 c	90.8 b	86.0 a	29.0 a	30.1 a
		14	235.6 c	180.4 d	182.1 ab	141.7 a	42.8 b	25.6 c	93.1 a	84.1 a	28.9 a	30.1 a
		Mean	258.2	210.0	175.2	130.2	45.0	27.1	92.0	84.0	29.2	30.2
	Nei6you107	27	241.2 b	231.5 a	198.6 a	122.3 b	47.8 a	28.2 a	94.1 a	83.2 a	28.1 a	29.9 ab
		23	287.0 a	223.2 a	158.0 c	127.0 b	45.2 ab	28.3 a	93.6 a	83.6 a	28.5 a	30.4 a
		18	243.5 b	194.4 b	176.4 b	143.6 a	42.9 bc	27.8 a	93.1 a	84.8 a	29.0 a	30 ab
		14	249.2 b	174.4 b	168.7 bc	149.8 a	41.6 c	26.1 a	95.6 a	83.7 a	28.7 a	29.6 b
		Mean	255.2	205.9	175.4	135.7	44.4	27.6	94.1	83.8	28.6	30.0
2021	Deyou4727	27	286.4 a	198.4 a	166.8 a	149.1 a	47.7 a	29.5 a	94.7 a	85.5 a	30.8 a	30.1 a
		23	265.9 ab	176.0 b	173.9 a	150.3 a	46.1 ab	26.4 b	93.8 a	86.5 a	30.3 a	30.5 a
		18	261.6 ab	169.8 bc	169.5 a	147.1 a	44.2 bc	25.0 c	95.3 a	88.1 a	30.2 a	30.2 a
		14	245.8 b	152.2 c	175.2 a	157.3 a	42.8 c	23.9 c	92.7 a	89.5 a	30.0 a	30.4 a
		Mean	264.9	174.1	171.3	150.9	45.2	26.2	94.1	87.4	30.3	30.3
	Luyou727	27	286.4 a	221.0 a	228.7 a	146.9 b	65.5 a	32.5 a	88.8 b	90.8 a	26.7 a	27.3 a
		23	287.0 a	172.2 b	206.2 a	186.5 a	58.9 b	32.0 a	91.9 ab	89.8 a	26.4 a	27.1 a
		18	278.1 a	172.9 b	209.9 a	177.0 a	57.9 b	30.6 a	90.6 ab	90.5 a	26.8 a	27.4 a
		14	270.6 a	166.8 b	204.2 a	180.1 a	55.3 b	30.0 a	92.4 a	90.3 a	26.9 a	26.8 a
		Mean	280.5	183.2	212.3	172.6	59.4	31.3	90.9	90.4	26.7	27.2
	Nei6you9	27	315.7 a	212.0 a	154.6 b	151.1 a	48.7 a	32.0 a	94.0 a	84.3 a	30.8 a	29.1 ab
		23	275.5 ab	183.7 ab	174.9 ab	162.5 a	48.2 a	29.8 ab	92.5 a	86.2 a	30.5 a	29.3 a
		18	257.0 b	169.8 b	186.6 a	165.4 a	47.8 a	28.0 b	93.2 a	85.3 a	30.1 a	28.3 b
		14	265.0 b	172.5 b	170.7 ab	159.5 a	45.2 a	27.3 b	93.5 a	85.7 a	30.6 a	28.3 b
		Mean	278.3	184.5	171.7	159.6	47.4	29.3	93.3	85.4	30.5	28.8
	Nei6you107	27	322.4 a	193.9 a	173.1 b	162.7 b	55.2 a	31.5 a	94.2 a	88.1 a	29.4 a	29.2 a
		23	277.4 b	185.6 a	185.4 b	162.4 b	51.1 b	30.1 ab	94.0 a	87.3 a	29.4 a	29.2 a
		18	234.5 c	153.3 b	209.9 a	183.3 a	49.2 b	28.0 ab	95.3 a	87.5 a	29.0 a	28.6 a
		14	242.4 c	148.8 b	179.8 b	181.1 a	43.6 c	26.9 b	94.2 a	85.9 a	29.1 a	28.6 a
		Mean	269.2	170.4	187.1	172.4	49.8	29.1	94.4	87.2	29.2	28.9

Note: Within the column for each cultivar, means of plant densities followed by different letters were significantly different according to LSD at *p* = 0.05.

**Table 3 plants-14-00660-t003:** Biomass production of super hybrid rice varieties grown under different plant densities in Hanyuan County and Luxian County of Sichuan Province in a field experiment from 2020 to 2021.

Year	Variety	Plant Density (Hills m^−2^)	Biomass Production (g m^−2^)						Harvest Index (%)	
			Pre-Heading		Post-Heading		Maturity Stage			
			Hanyuan	Luxian	Hanyuan	Luxian	Hanyuan	Luxian	Hanyuan	Luxian
2020	Deyou4727	27	1461.8 a	1160.6 a	708.5 a	346.9 a	2170.3 a	1507.5 a	55.7 ab	49.5 a
		23	1484.8 a	978.1 ab	513.2 b	404.3 a	1998.1 b	1382.3 b	57.6 a	48.2 a
		18	1472.2 a	1018.8 ab	515.9 b	323.9 a	1988.1 b	1342.7 b	55.3 b	49.1 a
		14	1451.8 a	882.1 b	503.0 b	390.4 a	1954.9 b	1272.6 c	55.8 ab	49.3 a
		Mean	1467.7	1009.9	560.2	366.4	2027.9	1376.3	56.1	49.0
	Luyou727	27	1485.2 b	1119.7 a	751.6 a	349.3 a	2236.8 a	1469.0 a	56.2 b	49.0 b
		23	1553.5 ab	1065.4 a	697.4 ab	419.4 a	2250.8 a	1484.8 a	57.6 a	51.6 a
		18	1589.8 a	1066.5 a	497.1 c	327.3 a	2086.9 b	1393.8 b	57.8 a	50.1 ab
		14	1379.1 c	1084.2 a	602.3 bc	290.3 a	1981.5 b	1374.5 b	53.7 c	50.7 ab
		Mean	1501.9	1084.0	637.1	346.6	2139.0	1430.5	56.3	50.3
	Nei6you9	27	1453.4 a	1074.0 a	638.1 b	516.6 a	2091.5 ab	1590.5 a	58.1 a	46.9 b
		23	1422.3 ab	953.4 b	821.2 a	474.0 ab	2243.5 a	1427.3 b	56.2 ab	47.7 b
		18	1480.5 a	1043.1 a	764.0 ab	309.6 c	2244.5 a	1352.7 c	53.6 b	49.8 a
		14	1318.9 b	936.7 b	745.0 ab	408.6 b	2063.9 b	1345.4 c	55.9 ab	48.0 b
		Mean	1418.8	1001.8	742.1	427.2	2160.9	1429.0	55.9	48.1
	Nei6you107	27	1370.2 a	1018.5 a	798.7 ab	395.6 a	2168.8 a	1414.1 b	58.3 a	49.6 a
		23	1391.3 a	1036.1 a	750.3 ab	468.1 a	2141.6 ab	1504.2 a	56.4 ab	47.9 a
		18	1367.9 a	960.6 a	642.1 b	478.3 a	2010.0 b	1438.9 b	57.5 ab	49.2 a
		14	1228.3 b	930.4 a	849.4 a	414.6 a	2077.7 ab	1345.0 c	55.1 b	48.0 a
		Mean	1339.4	986.4	760.1	439.1	2099.5	1425.6	56.8	48.7
2021	Deyou4727	27	1726.0 a	1025.8 a	812.0 a	423.6 a	2538.0 a	1449.4 a	54.8 a	52.3 a
		23	1663.2 ab	1040.5 a	754.9 a	311.7 a	2418.0 b	1352.2 ab	54.2 a	51.6 a
		18	1733.7 a	998.7 ab	591.6 b	309.1 a	2325.2 c	1307.7 ab	54.5 a	50.9 a
		14	1615.3 b	863.1 b	720.2 a	399.0 a	2335.5 c	1262.1 b	51.0 b	51.5 a
		Mean	1684.6	982.0	719.7	360.8	2404.2	1342.9	53.6	51.6
	Luyou727	27	1654.6 a	1057.3 a	980.7 a	501.2 a	2635.4 a	1558.6 a	58.8 a	51.6 a
		23	1630.8 a	994.4 ab	799.3 b	504.6 a	2430.1 b	1499.0 a	58.8 a	51.9 a
		18	1504.3 a	1091.7 a	1022.0 a	384 a	2526.4 ab	1475.7 a	55.7 b	51.4 a
		14	1503.8 a	925.5 b	946.9 ab	512.9 a	2450.7 b	1438.3 a	55.9 b	50.6 a
		Mean	1573.4	1017.2	937.2	475.7	2510.7	1492.9	57.3	51.4
	Nei6you9	27	1684.9 a	1063.5 a	987.9 a	447.8 a	2672.8 a	1511.3 a	52.7 a	52.0 a
		23	1536.0 b	1142.7 a	996.7 a	318.7 a	2532.7 b	1461.4 a	53.7 a	51.4 a
		18	1729.5 a	921.5 b	801.3 b	404.6 a	2530.8 b	1326.2 b	52.9 a	50.9 a
		14	1526.7 b	877.0 b	972.7 a	417.4 a	2499.4 b	1294.4 b	51.8 a	51.0 a
		Mean	1619.3	1001.2	939.6	397.1	2558.9	1398.3	52.8	51.3
	Nei6you107	27	1710.8 a	1009.8 a	981.8 a	530.6 a	2692.5 a	1540.5 a	56.8 a	52.6 a
		23	1573.9 ab	946.1 a	921.9 ab	533.1 a	2495.8 b	1479.1 ab	56.7 a	51.7 ab
		18	1469.2 b	1038.9 a	895.8 ab	328.6 a	2365.0 c	1367.5 bc	57.5 a	51.2 b
		14	1478.5 b	863.9 a	757.3 b	442.4 a	2235.8 d	1306.4 c	53.3 b	50.6 b
		Mean	1558.1	964.7	889.2	458.7	2447.3	1423.4	56.1	51.5

Note: Within the column for each cultivar, means of plant densities followed by different letters were significantly different according to LSD at *p* = 0.05.

**Table 4 plants-14-00660-t004:** Crop growth rate (CGR, g m^−2^ d^−1^) and apparent radiation use efficiency (RUE, g MJ^−1^) of super hybrid rice varieties grown under different plant densities in Hanyuan County and Luxian County of Sichuan Province in a field experiment from 2020 to 2021.

Year	Variety	Plant Density (Hills m^−2^)	Crop Growth Rate (g m^−2^ d^−1^)				Apparent Radiation Use Efficiency (g MJ^−2^)			
			Pre-Heading		Post-Heading		Pre-Heading		Post-Heading	
			Hanyuan	Luxian	Hanyuan	Luxian	Hanyuan	Luxian	Hanyuan	Luxian
2020	Deyou4727	27	15.4 a	12.6 a	17.3 a	11.2 a	0.80 a	0.71 a	1.05 a	0.64 a
		23	15.6 a	10.6 ab	12.5 b	13.0 a	0.81 a	0.60 ab	0.76 b	0.74 a
		18	15.5 a	11.1 ab	12.6 b	10.4 a	0.80 a	0.62 ab	0.76 b	0.60 a
		14	15.3 a	9.6 b	12.3 b	12.6 a	0.79 a	0.54 b	0.74 b	0.72 a
		Mean	15.4	11.0	13.7	11.8	0.80	0.62	0.83	0.67
	Luyou727	27	15.0 b	11.8 a	16.3 a	10.6 a	0.78 b	0.66 a	1.04 a	0.60 a
		23	15.7 ab	11.2 a	15.2 ab	12.7 a	0.82 ab	0.63 a	0.97 ab	0.72 a
		18	16.1 a	11.2 a	10.8 c	9.9 a	0.84 a	0.63 a	0.69 c	0.56 a
		14	13.9 c	11.4 a	13.1 bc	8.8 a	0.73 c	0.64 a	0.84 bc	0.50 a
		Mean	15.2	11.4	13.9	10.5	0.79	0.64	0.88	0.59
	Nei6you9	27	15.0 a	11.5 a	15.2 b	16.1 a	0.78 a	0.65 a	0.92 b	0.92 a
		23	14.7 ab	10.3 b	19.6 a	14.8 ab	0.77 ab	0.58 b	1.19 a	0.84 ab
		18	15.3 a	11.2 a	18.2 ab	9.7 c	0.80 a	0.63 a	1.11 ab	0.55 c
		14	13.6 b	10.1 b	17.7 ab	12.8 b	0.71 b	0.57 b	1.08 ab	0.73 b
		Mean	14.6	10.8	17.7	13.4	0.77	0.60	1.07	0.76
	Nei6you107	27	14.0 a	10.8 a	18.6 ab	12.4 a	0.73 a	0.61 a	1.15 ab	0.70 a
		23	14.2 a	11.0 a	17.4 ab	14.6 a	0.74 a	0.62 a	1.08 ab	0.83 a
		18	14.0 a	10.2 a	14.9 b	14.9 a	0.73 a	0.57 a	0.93 b	0.85 a
		14	12.5 b	9.9 a	19.8 a	13.0 a	0.65 b	0.55 a	1.23 a	0.74 a
		Mean	13.7	10.5	17.7	13.7	0.71	0.59	1.10	0.78
2021	Deyou4727	27	18.2 a	11.3 a	20.3 a	14.1 a	0.94 a	0.67 a	1.23 a	0.75 a
		23	17.5 ab	11.4 a	18.9 a	10.4 a	0.90 ab	0.68 a	1.14 a	0.55 a
		18	18.2 a	11.0 ab	14.8 b	10.3 a	0.94 a	0.65 ab	0.90 b	0.55 a
		14	17.0 b	9.5 b	18.0 a	13.3 a	0.88 b	0.56 b	1.09 a	0.71 a
		Mean	17.7	10.8	18.0	12.0	0.91	0.64	1.09	0.64
	Luyou727	27	16.9 a	11.2 a	22.3 a	15.7 a	0.87 a	0.67 a	1.42 a	0.85 a
		23	16.6 a	10.6 ab	18.2 b	15.8 a	0.85 a	0.63 ab	1.16 b	0.85 a
		18	15.4 a	11.6 a	23.2 a	12.0 a	0.79 a	0.69 a	1.48 a	0.65 a
		14	15.3 a	9.8 b	21.5 ab	16.0 a	0.79 a	0.59 b	1.37 ab	0.87 a
		Mean	16.1	10.8	21.3	14.9	0.82	0.65	1.36	0.80
	Nei6you9	27	17.6 a	11.6 a	24.1 a	14.4 a	0.90 a	0.69 a	1.48 a	0.77 a
		23	16.0 b	12.4 a	24.3 a	10.3 a	0.82 b	0.74 a	1.49 a	0.55 a
		18	18.0 a	10.0 b	19.5 b	13.1 a	0.93 a	0.60 b	1.20 b	0.70 a
		14	15.9 b	9.5 b	23.7 a	13.5 a	0.82 b	0.57 b	1.46 a	0.72 a
		Mean	16.9	10.9	22.9	12.8	0.87	0.65	1.41	0.68
	Nei6you107	27	17.6 a	10.9 a	23.4 a	17.1 a	0.91 a	0.65 a	1.50 a	0.91 a
		23	16.2 ab	10.2 a	21.9 ab	17.2 a	0.83 ab	0.61 a	1.41 ab	0.92 a
		18	15.1 b	11.2 a	21.3 ab	10.6 a	0.78 b	0.67 a	1.37 ab	0.57 a
		14	15.2 b	9.3 a	18.0 b	14.3 a	0.78 b	0.55 a	1.16 b	0.76 a
		Mean	16.1	10.4	21.2	14.8	0.82	0.62	1.36	0.79

Note: Within the column for each cultivar, means of plant densities followed by the different letters were significantly different according to LSD at *p* = 0.05.

**Table 5 plants-14-00660-t005:** Grain yield of four super hybrid rice varieties grown in Hanyuan County and Luxian County of Sichuan Province in a pot experiment in 2020 and 2021.

Location	Soil ^#^	Variety	Grain Yield per Hill (g)	
			2020	2021
Hanyuan	Hanyuan	Deyou4727	35.95 b	29.52 a
		Luyou727	38.13 a	30.33 a
		Nei6you9	36.98 ab	30.13 a
		Nei6you107	35.12 b	29.89 a
		Mean	36.55	29.97
	Luxian	Deyou4727	26.67 a	28.15 a
		Luyou727	28.17 a	30.65 a
		Nei6you9	26.65 a	28.56 a
		Nei6you107	27.68 a	29.84 a
		Mean	27.29	29.30
Luxian	Hanyuan	Deyou4727	20.98 a	25.08 ab
		Luyou727	23.43 a	29.86 a
		Nei6you9	20.52 a	24.09 b
		Nei6you107	22.08 a	29.16 a
		Mean	21.75	27.05
	Luxian	Deyou4727	19.48 a	22.59 ab
		Luyou727	22.14 a	20.05 bc
		Nei6you9	20.81 a	24.26 a
		Nei6you107	21.38 a	17.95 c
		Mean	20.95	21.21
Analysis of variance (ANOVA)				
Location (L)			**	**
Soil (S)			**	**
Variety (V)			ns	ns
L × S			**	**
L × V			ns	ns
S × V			ns	ns
L × S × V			ns	**

Note: Within the column for each soil, means of cultivars followed by different letters were significantly different according to LSD at *p* = 0.05. **^#^** The soil used in the pot experiments was collected from the top 25 cm layer of fields at the experimental stations of Hanyuan County and Luxian County, Sichuan Province. ** Significant at the 0.01 level based on analysis of variance. ns denotes non-significance based on analysis of variance.

**Table 6 plants-14-00660-t006:** Yield components, biomass, and harvest index of four super hybrid rice varieties grown in Hanyuan County and Luxian County of Sichuan Province in a pot experiment in 2020 and 2021.

Year	Location	Soil ^#^	Variety	Panicles per m^2^	Spikelets per Panicle	Grain Filling (%)	Grain Weight (mg)	Biomass (g Hill^−1^)	Harvest Index (%)
2020	Hanyuan	Hanyuan	Deyou4727	10.8 a	119.6 b	93.9 a	29.9 ab	71.1 ab	51.4 a
			Luyou727	10.3 a	154.9 a	89.8 a	26.7 c	75.5 a	49.8 a
			Nei6you9	9.7 a	133.0 b	93.2 a	31.2 a	60.0 b	51.7 a
			Nei6you107	10.5 a	131.3 b	89.7 a	29.0 b	69.3 ab	50.0 a
			Mean	10.3	134.7	91.6	29.2	69.0	50.7
		Luxian	Deyou4727	10.3 a	96.5 a	91.0 ab	30.8 a	56.1 a	47.7 a
			Luyou727	10.3 a	114.6 a	86.9 b	27.5 c	56.5 a	49.8 a
			Nei6you9	9.3 a	112.2 a	88.3 ab	29.8 b	46.1 b	47.0 a
			Nei6you107	9.7 a	103.1 a	92.2 a	29.8 b	55.4 ab	46.9 a
			Mean	9.9	106.6	89.6	29.4	53.5	47.8
	Luxian	Hanyuan	Deyou4727	7.5 a	147.9 ab	89.9 a	29.5 a	42.3 a	49.4 ab
			Luyou727	7.7 a	156.4 a	88.3 ab	25.6 c	45.6 a	51.4 a
			Nei6you9	6.8 a	125.9 b	85.4 b	28.5 b	44.2 a	46.3 b
			Nei6you107	7.1 a	130.0 ab	86.9 ab	28.5 b	44.1 a	50.2 ab
			Mean	7.3	140.0	87.6	28.0	44.1	49.3
		Luxian	Deyou4727	6.8 b	106.2 b	90.6 a	30.4 a	40.1 a	48.8 a
			Luyou727	6.5 b	144.7 a	90.9 a	26.1 d	42.9 a	51.6 a
			Nei6you9	8.0 a	109.4 b	85.2 b	28.3 c	43.3 a	47.9 a
			Nei6you107	7.2 ab	117.4 b	88.2 ab	29.1 b	43.4 a	49.2 a
			Mean	7.1	119.4	88.7	28.4	42.4	49.4
2021	Hanyuan	Hanyuan	Deyou4727	10.6 ab	115.6 a	88.7 a	30.3 a	54.9 a	53.9 b
			Luyou727	11.3 a	121.6 a	87.4 a	27.9 b	53.0 ab	57.2 ab
			Nei6you9	9.7 bc	123.3 a	93.0 a	30.6 a	53.7 a	56.3 ab
			Nei6you107	9.2 c	134.0 a	92.4 a	30.4 a	49.9 b	59.8 a
			Mean	10.2	123.6	90.4	29.8	52.9	56.8
		Luxian	Deyou4727	10.1 ab	99.6 a	93.5 a	30.3 a	51.2 b	55.3 a
			Luyou727	11.0 a	115.9 a	89.2 b	27.6 b	56.6 a	55.1 a
			Nei6you9	9.3 b	107.0 a	94.6 a	30.5 a	51.2 b	55.4 a
			Nei6you107	9.0 b	114.0 a	94.9 a	30.9 a	51.8 b	57.6 a
			Mean	9.9	109.1	93.0	29.8	52.7	55.8
	Luxian	Hanyuan	Deyou4727	9.0 a	116.4 b	93.4 a	29.0 a	56.5 ab	50.8 a
			Luyou727	9.3 a	138.2 ab	87.8 b	26.3 c	60.1 a	49.8 a
			Nei6you9	6.8 b	147.4 a	85.7 b	27.7 b	47.6 b	50.4 a
			Nei6you107	8.8 a	138.0 ab	86.3 b	28.0 b	59.6 ab	49.2 a
			Mean	8.5	135.0	88.3	27.7	56.0	50.0
		Luxian	Deyou4727	7.9 a	102.5 b	92.7 a	30.4 a	44.8 a	50.7 a
			Luyou727	6.6 bc	132.1 a	89.1 b	26.6 d	38.8 ab	51.9 a
			Nei6you9	7.4 ab	130.9 a	89.8 ab	28.0 c	46.7 a	51.9 a
			Nei6you107	6.0 c	115.7 ab	89.9 ab	28.9 b	34.1 b	52.7 a
			Mean	7.0	120.3	90.4	28.5	41.1	51.8

Note: within the column for each soil, means of cultivars followed by different letters were significantly different according to LSD at *p* = 0.05. **^#^** The soil used in the pot experiments was collected from the top 25 cm layer of fields at the experimental stations of Hanyuan County and Luxian County, Sichuan Province.

## Data Availability

Data are contained within the article.

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
