# Peer review of "Yield Performance of Super Hybrid Rice Grown in Subtropical Environments at a Similar Latitude but Different Altitudes in Southwest China"

_plants, 2025, doi:10.3390/plants14050660_

Round 1
Reviewer 1 Report
Comments and Suggestions for Authors
Would the difference between air pressure and CO2 concentration at 1000 m (Hanyuan) and 300 m (Luxian) be a factor affecting any of traits studied?
Author Response
Dear reviewer and editors,
We thank for the reviewer’s valuable comments on the manuscript entitled “Yield performance of super hybrid rice grown on subtropical environments at similar latitude but different altitudes” (ID:plants-3304475). All comments are highly valuable for improving the manuscript. We have completely revised the manuscript in accordance with the reviewers’ comments. All changes are marked in red in the manuscript, and our point-by-point responses are listed below.
Comment: Would the difference between air pressure and CO2 concentration at 1000 m (Hanyuan) and 300 m (Luxian) be a factor affecting any of traits studied?
Response: Excellent point. Unfortunately, we do not collect the parameters of air pressure and CO2 concentration in the present study. In general, the temperature and solar radiation intensity are the two most important factors for limiting rice grain yield. A large differences in maximum and minimum temperatures and solar radiation existed between two locations, and Hanyuan exhibited superior light conditions and temperatures compared to Luxian. Grain yield was significantly correlated with maximum and minimum temperatures and solar radiation. This finding indicated that the maximum and minimum temperatures can explained the rice yield difference in two locations. This suggestion will be incorporated in future research.
The revised manuscript has been submitted to your journal. We look forward to your positive responses. Thank you very much and best regards.
Sincerely yours,
Peng jiang
Rice and Sorghum Research Institute, Sichuan Academy of Agricultural Sciences
Deyang 618000, China
FuXian Xu
Rice and Sorghum Research Institute, Sichuan Academy of Agricultural Sciences
Deyang 618000, China
Reviewer 2 Report
Comments and Suggestions for Authors
Too many of the investigated parameters are presented in tabular form, making it almost impossible to follow the work's aim and the results obtained. Consequently, the conclusion cannot be observed. Statistical procedures should be applied in such cases to extract valuable findings.
The complexity of the discussion and conclusion in this work arises from the multitude of parameters, some of which appear to be redundant or superfluous. Take, for instance, the data in Table 1. As a consequence, the parameter naming within the Discussion subsection is hard to track and connect. I recommend employing a statistical method like Principal Component Analysis (PCA) to evaluate the relevance of the parameters involved. This method could make the ensuing discussions and conclusions more comprehensible. The goal should not be to amass an overwhelming quantity of data without meaningful context. Additionally, the tables are not easily readable; thus, it would be prudent to include them as appendices. In contrast, data that has been statistically analyzed and derived from these tables should be presented within the main text. I am convinced that this would greatly clarify the discussion and conclusions.
Author Response
Dear reviewer and editors,
We thank for the reviewer’s valuable comments on the manuscript entitled “Yield performance of super hybrid rice grown on subtropical environments at similar latitude but different altitudes” (ID:plants-3304475). All comments are highly valuable for improving the manuscript. We have completely revised the manuscript in accordance with the reviewers’ comments. All changes are marked in red in the manuscript, and our point-by-point responses are listed below.
Comment 1: Too many of the investigated parameters are presented in tabular form, making it almost impossible to follow the work's aim and the results obtained. Consequently, the conclusion cannot be observed. Statistical procedures should be applied in such cases to extract valuable findings.
Response 1: In the present study, statistical analyses were conducted using the Statistix 8 software package with an analysis of variance (ANOVA) approach. In field experiments, the ANOVA model included the following factors: location (L), plant density (P), variety (V), and interactions between these factors, specifically L×P, L×V, and P×V, and the three-factor interaction of L×P×V. Location (L), variety (V) and plant density (P) all had a significant impact on the grain yield of super hybrid rice varieties in field experiments. Based on the results of field experiments, we could clarify that the relationship of grain yield and yield attributes with altitude. In pot experiments, the ANOVA model similarly included location (L), soil fertility (S), and variety (V), as well as the two-factor interactions L×S, L×V, and S×V, and the three-factor interaction L×S×V. Grain yield was significantly influenced by location (L) and soil type (S), and a significant interaction was observed between location and soil fertility in pot experiments. Based on the results of pot experiments, we could clarify that the relationship of grain yield and yield attributes with soil fertility. Considering the results from both the paddy field and pot experiments, we could clarify that the relationship of grain yield with altitude and soil fertility. So, the statistical approach is appropriate.
Comment 2: The complexity of the discussion and conclusion in this work arises from the multitude of parameters, some of which appear to be redundant or superfluous. Take, for instance, the data in Table 1. As a consequence, the parameter naming within the Discussion subsection is hard to track and connect. I recommend employing a statistical method like Principal Component Analysis (PCA) to evaluate the relevance of the parameters involved. This method could make the ensuing discussions and conclusions more comprehensible.The goal should not be to amass an overwhelming quantity of data without meaningful context. Additionally, the tables are not easily readable; thus, it would be prudent to include them as appendices. In contrast, data that has been statistically analyzed and derived from these tables should be presented within the main text. I am convinced that this would greatly clarify the discussion and conclusions.
Response 2: The growth duration of hybrid rice and its corresponding temperatures and solar radiation were showed in the table 1. These data in table 1 are very important to explain the differences in rice yield and yield traits between two location. Which is not redundant or superfluous. We have been employed the statistical method like correlation analysis between indexes (including the yield traits and climatic conditions) and grain yield of four hybrid rice varieties to evaluate the relevance of the parameters involved. A large differences in grain yield and yield traits of super hybrid rice at various ecological sites are primary related to soil fertility and climatic conditions. This finding indicated that the soil fertility and climatic conditions could explain the differences in grain yield and yield traits, and Hanyuan exhibited superior soil fertility and climactic conditions compared to Luxian. Therefore, it is not necessary to use statistical method like Principal Component Analysis (PCA) again to evaluate the relevance of the parameters.
The revised manuscript has been submitted to your journal. We look forward to your positive responses. Thank you very much and best regards.
Sincerely yours,
Peng jiang
Rice and Sorghum Research Institute, Sichuan Academy of Agricultural Sciences
Deyang 618000, China
FuXian Xu
Rice and Sorghum Research Institute, Sichuan Academy of Agricultural Sciences
Deyang 618000, China
Reviewer 3 Report
Comments and Suggestions for Authors
In this study, the authors compared growth and yield performance of hybrid rice varieties grown in two fields at different altitudes but the same latitude. The experimental design is novel, and the dataset is comprehensive. However, the authors failed to develop well-structured arguments from their results.
The manuscript requires major revision before publication consideration. While the research design has merit and the data are valuable, both the presentation and analysis require substantial improvement to effectively communicate the findings.
The manuscript requires revision to address the following concerns.
Major concerns:
1. Rationale
The authors failed to justify their study objectives and experimental design clearly. The manuscript lacks appropriate hypotheses, which leads to unfocused discussion of the experimental results. While the authors appear to attempt elucidating the mechanisms behind varied yield performances of hybrid rice reported in previous studies, they failed to discuss their data sufficiently from this perspective. The authors attributed differences in plant performance between the two locations to soil fertility, yet provided no detailed soil fertility analysis to support this claim. Additionally, in the Abstract and Conclusion sections, the authors suggested they identified physiological characteristics that could break the yield ceiling in hybrid rice varieties. However, their data interpretation does not fully support these arguments. The authors should restructure their manuscript by incorporating clear, testable hypotheses.
2. Statistical Analysis
The statistical approach is inappropriate for the experimental design. Given the nested structure of the data, the analysis should employ either nested ANOVA or generalized linear mixed model. The results should be presented with appropriate error terms and significance levels.
3. Data Presentation
All figures and tables require revision to be standalone. Current issues include 1) undefined abbreviations, 2) incomplete legends, and 3) absence of marginal means.
4. Soil Analysis
The authors' emphasis on soil fertility requires more comprehensive analysis. The authors should 1) present soil characteristics in a detailed table, 2) describe soil sampling methodology and timing clearly, 3) document temporal changes in soil parameters, and 4) analyze relationships between soil properties and yield components.
Specific comments:
Introduction:
- L78-L82: Provide evidence to support these claims
- L97-L99: Explain the relationship between yield components and environmental conditions
- L119-L120: Include quantitative analysis of published data inconsistencies
Methods:
- L521-L530: Present soil characteristics in a comprehensive table
- L576-L577: Specify soil sampling timeline and frequency
Results and Discussion:
- L193-L195: Revise for clarity and accuracy
- L365-L366: Verify yield comparison data
- L372-L373: Provide evidence for this statement
- L615-L616: Support claim with experimental data
The Introduction and Discussion sections contain redundant and irrelevant content that should be removed.
The Abstract requires revision to eliminate ambiguous statements such as "provide valuable insights for enhancing super hybrid rice productivity" and "superior light conditions and temperatures compared to Luxian." The revised Abstract should clearly present the specific findings and conclusions supported by the data.
Author Response
Dear reviewer and editors,
We thank for the reviewer’s valuable comments on the manuscript entitled “Yield performance of super hybrid rice grown on subtropical environments at similar latitude but different altitudes” (ID:plants-3304475). All comments are highly valuable for improving the manuscript. We have completely revised the manuscript in accordance with the reviewers’ comments. All changes are marked in red in the manuscript, and our point-by-point responses are listed below.
Comment 1: The authors failed to justify their study objectives and experimental design clearly. The manuscript lacks appropriate hypotheses, which leads to unfocused discussion of the experimental results. While the authors appear to attempt elucidating the mechanisms behind varied yield performances of hybrid rice reported in previous studies, they failed to discuss their data sufficiently from this perspective. The authors attributed differences in plant performance between the two locations to soil fertility, yet provided no detailed soil fertility analysis to support this claim. Additionally, in the Abstract and Conclusion sections, the authors suggested they identified physiological characteristics that could break the yield ceiling in hybrid rice varieties. However, their data interpretation does not fully support these arguments. The authors should restructure their manuscript by incorporating clear, testable hypotheses.
Response 1: Agree. We have been revised the manuscript according to the reviewer’s suggestion, and the add the hypotheses in Line 131-136. The abstract and conclusion have been revised. Line30-33, 37-39 and 41-43; Line392-394, 410-413; Line620, 632-634.
Comment 2: The statistical approach is inappropriate for the experimental design. Given the nested structure of the data, the analysis should employ either nested ANOVA or generalized linear mixed model. The results should be presented with appropriate error terms and significance levels.
Response 2: In our manuscript, data analysis were carried out with the the Statistix 8 software package (Analytical Software, Tallahassee, Florida, USA). For field experiments, the statistical model of the analysis of variance (ANOVA) included location, plant density, variety, and the interactions between these factors. Clarifying the effects of location, plant density, variety, and the interactions between these factors on grain yield and yield attributes of super hybrid rice. For the pot experiments, the ANOVA model similarly included location, soil, and variety, as well as the two-factor interactions L×S, L×V, and S×V, and the three-factor interaction L×S×V. Clarifying the effects of location, soil, and variety and the interactions between these factors on grain yield and yield attributes of super hybrid rice. Correlation plot analysis was used to evaluate the relationships between grain yield, yield attributes and climatic factors (Origin 2024, OriginLab Corp., Northampton, MA, USA). Based on the results from both field and pot experiments. it is easily to isolate the individual contributions of altitude, latitude and soil fertility to the variations in super high rice yields based on the findings of both the paddy field and pot experiments. So, the statistical approach is appropriate.
Comment 3: All figures and tables require revision to be standalone. Current issues include 1) undefined abbreviations, 2) incomplete legends, and 3) absence of marginal means.
Response 3: We have been revised the manuscript according to the reviewer’s suggestion. Table 1, 2, 3, 4, 5 and 6.
Comment 4: The authors' emphasis on soil fertility requires more comprehensive analysis. The authors should 1) present soil characteristics in a detailed table, 2) describe soil sampling methodology and timing clearly, 3) document temporal changes in soil parameters, and 4) analyze relationships between soil properties and yield components.
Response 4: The soil characteristics were showed in table 9. Soil testing was performed on samples taken from the upper 20 cm layer before transplanting the rice seedlings in 2020. In this study, Hanyuan exhibited higher organic matter, total nitrogen, total phosphorus, total potassium, available nitrogen, available potassium compared to Luxian. Line 392-394, 409-411; Line 524-526.
Comment 5: L78-L82: Provide evidence to support these claims
Response 5: We have been deleted the sentences.
Comment 6: L97-L99: Explain the relationship between yield components and environmental conditions
Response 6: Increasing plant density is an effective approach to increasing panicles per unit land area that achieving high grain yields in southern China.
Comment 7: L119-L120: Include quantitative analysis of published data inconsistencies
Response 7: We have been revised the manuscript according to the reviewer’s suggestion. Line112-116.
Comment 8: L521-L530: Present soil characteristics in a comprehensive table
Response 8: The soil characteristics of the paddy field in two location were showed in Table 9.
Comment 9: L576-L577: Specify soil sampling timeline and frequency
Response 9: Soil testing was performed on samples taken from the upper 20 cm layer before transplanting the rice seedlings in 2020.
Comment 10: L193-L195: Revise for clarity and accuracy
Response 10: We have been revised the manuscript according to the reviewer’s suggestion. Line188-189
Comment 11: L365-L366: Verify yield comparison data
Response 11: We have checked the comparison data, the highest grain yield were 15.2, 14.1, 16.5 and 15.8 t ha-1 in the reference 8-10, 13, respectively. The highest grain yield in the present study is higher than those reported in earlier studies at high-altitude sites. Line371-372
Comment 12: L372-L373: Provide evidence for this statement
Response 12: The climatic conditions were showed in Table 1. We have been revised the manuscript according to the reviewer’s suggestion. Line 375-384.
Comment 13: L615-L616: Support claim with experimental data
Response 13:We have been revised the manuscript according to the reviewer’s suggestion. Line615-616
Comment 14: The Introduction and Discussion sections contain redundant and irrelevant content that should be removed.
Response 14: We have been revised the manuscript according to the reviewer’s suggestion.The Introduction: Line80-81, 99-103, 112-116, 131-136; Th Discussion: Line365-368, 371-372, 375-384, 386-388, 392-394, 395-399, 409-413,462-466, 510-517.
Comment 15: The Abstract requires revision to eliminate ambiguous statements such as "provide valuable insights for enhancing super hybrid rice productivity" and "superior light conditions and temperatures compared to Luxian." The revised Abstract should clearly present the specific findings and conclusions supported by the data.
Response 15: We have been revised the manuscript according to the reviewer’s suggestion. Line30-33, 38-39, 41-43
The revised manuscript has been submitted to your journal. We look forward to your positive responses. Thank you very much and best regards.
Sincerely yours,
Peng jiang
Rice and Sorghum Research Institute, Sichuan Academy of Agricultural Sciences
Deyang 618000, China
FuXian Xu
Rice and Sorghum Research Institute, Sichuan Academy of Agricultural Sciences
Deyang 618000, China
Reviewer 4 Report
Comments and Suggestions for Authors
I understand that the article has meaningful data to analyze the improvement of super hybrid rice yields by the difference in altitude and soil nutrients in Sichuan Province, China.
However, there are some improvements to the manuscript needed as attached:

Author Response
Dear reviewer and editors,
We thank for the reviewer’s valuable comments on the manuscript entitled “Yield performance of super hybrid rice grown on subtropical environments at similar latitude but different altitudes” (ID:plants-3304475). All comments are highly valuable for improving the manuscript. We have completely revised the manuscript in accordance with the reviewers’ comments. All changes are marked in red in the manuscript, and our point-by-point responses are listed below.
Comment 1. Title: I don’t think that all subtropical environmental areas fit the same results for the effect of altitude on super hybrid rice biomass and grain yields. Therefore, I recommend that the location of the present experiment “in Sichuan Province, China” should better add at the end of the paper’s title.
Response 1: We have been changed the title as followed: Yield performance of super hybrid rice grown on subtropical environments at similar latitude but different altitudes in Southwest China. Line 3.
Comment 2. Abstract: Hanyuan (High-altitude at 1,000 m above sea level, asl.) and Luxian (low-altitude, 300 m asl.) should better be added for the reader to easily understand the difference in altitude between two sites.
Response 2: We appreciate your suggestion. Due to the word limit of the abstract, we did not provide detailed latitude and longitude information for these two locations. The Materials and Methods section displays the detailed latitude and longitude of these two locations.
Comment 3. Discussion for meaning of difference in altitude: In the present manuscript, the difference in altitude is closely linked with the difference in cumulative solar radiation, where the high altitude site has higher in cumulative solar radiation. With the increase in altitude, the decrease in mean temperature (both maximum and minimum temperatures) is quite reasonable, however, it is not common that the increase in altitude should increase cumulative solar radiation. Therefore, the authors should ask for understanding beforehand in the positive correlation obtained between altitude and cumulative solar radiation in the examined two sites.
Response 3: The cumulative solar radiation was determined by solar radiation intensity and growth duration. In the present study, the higher cumulative solar radiation before heading in Hanyuan was associated with high solar radiation intensity and longer growth duration, while the higher cumulative solar radiation after heading observed at Hanyuan was linked to longer growth duration from heading to maturity.
Comment 4. From Table 2 to Table 5: The second column is “Plant density (hills per m2 )”. However, the density is expressed as “D1, D2, D3, D4”, which is difficult to understand the effect of plant density directly. Therefore, I recommend replacing D1, D2, D3, and D4 to 27.0, 23.0, 18.0, and 13.5 (hills per m2 ), which is quite easily understood for the effect of plant density.
Response 4:Agree. We have been revised the manuscript according to the reviewer’s suggestion. Table 2-5.
Comment 5. Discussion: The compound expression is difficult to catch the meaning such as, “the mean temperature before heading” and “cumulative solar radiation until heading” are, I think, both the pre-heading climatic factors, however the prepositions are different. Please unify the preposition, if there is no reason for using different words.
Response 5: Agree. We have been revised the manuscript according to the reviewer’s suggestion. Line 441, 443, 445, 458, 461, 476,
Comment 6. Effects of experimental years between 2020 and 2021 are not discussed well in the present manuscript. Especially, the grain yield in Hanyang were higher than in Luxian both in 2020 and 2021 across four varieties and all four plant densities for the field experiment as well as in the pot experiment for the combination of Location: Hanyuan to Soil: Luxian, Location: Luxian to both soils: Hanyuan and Luxian. However, there was the only exception for the combination of Location: Hanyuan to Soil: Hanyuan for the pot experiment. It is difficult to understand the rational reason for these responses in grain yield. Therefore, I recommend describing the experimental conditions to conduct the pot experiments at Hanyuan in 2020 or 2021 to understand these responses.
Response 6: Agree. We have been revised the manuscript according to the reviewer’s suggestion. Line 368-372, 375-384, 386-388, 392-394, 395-399, 409-413, 462-466, 510-517
The revised manuscript has been submitted to your journal. We look forward to your positive responses. Thank you very much and best regards.
Sincerely yours,
Peng jiang
Rice and Sorghum Research Institute, Sichuan Academy of Agricultural Sciences
Deyang 618000, China
FuXian Xu
Rice and Sorghum Research Institute, Sichuan Academy of Agricultural Sciences
Deyang 618000, China
Round 2
Reviewer 2 Report
Comments and Suggestions for Authors
Despite the corrections and additions made to the manuscript and attempts to clarify the presentation of results in extensive tables, I maintain that the research findings are still presented in an unreadable manner, making it difficult to follow the flow of the results. While many parameters are included, the number of plant individuals sampled and the specific area from which they were collected are not provided. Additionally, the results from pot experiments are included, which seems redundant. As a result, the conclusion drawn is overly simplistic.
Moreover, the order of the chapters in the revised manuscript is entirely disorganized. The Results chapter appears after the Introduction, followed by the Discussion chapter, then the Materials and Methods chapter, and finally the Conclusion.
Author Response
Dear reviewer and editors,
We thank for the reviewer’s valuable comments on the manuscript entitled “Yield performance of super hybrid rice grown on subtropical environments at similar latitude but different altitudes” (ID:plants-3304475). All comments are highly valuable for improving the manuscript. We have completely revised the manuscript in accordance with the reviewers’ comments. All changes are marked in red in the manuscript, and our point-by-point responses are listed below.
Comment 1: Despite the corrections and additions made to the manuscript and attempts to clarify the presentation of results in extensive tables, I maintain that the research findings are still presented in an unreadable manner, making it difficult to follow the flow of the results. While many parameters are included, the number of plant individuals sampled and the specific area from which they were collected are not provided. Additionally, the results from pot experiments are included, which seems redundant. As a result, the conclusion drawn is overly simplistic.
Response 1: We have provided the methods of plants samples and sample area. The objective of the pot experiment was to isolate the individual contributions of soil fertility and climatic conditions to the variations in super high rice yields. So, the pot experiments is not redundant. The conclusion have been improved. Line 620-627
Comment 2: Moreover, the order of the chapters in the revised manuscript is entirely disorganized. The Results chapter appears after the Introduction, followed by the Discussion chapter, then the Materials and Methods chapter, and finally the Conclusion.
Response 2: The order of the chapters in the revised manuscript is organized based on the requirements from Journal of Plants.
The revised manuscript has been submitted to your journal. We look forward to your positive responses. Thank you very much and best regards.
Sincerely yours,
Peng jiang
Rice and Sorghum Research Institute, Sichuan Academy of Agricultural Sciences
Deyang 618000, China
FuXian Xu
Rice and Sorghum Research Institute, Sichuan Academy of Agricultural Sciences
Deyang 618000, China

Reviewer 3 Report
Comments and Suggestions for Authors
The authors failed to address some of my major concerns
Author Response
Dear reviewer and editors,
We thank for the reviewer’s valuable comments on the manuscript entitled “Yield performance of super hybrid rice grown on subtropical environments at similar latitude but different altitudes” (ID:plants-3304475). All comments are highly valuable for improving the manuscript. We have completely revised the manuscript in accordance with the reviewers’ comments. All changes are marked in red in the manuscript, and our point-by-point responses are listed below.
Comment 1: The authors failed to justify their study objectives and experimental design clearly. The manuscript lacks appropriate hypotheses, which leads to unfocused discussion of the experimental results. While the authors appear to attempt elucidating the mechanisms behind varied yield performances of hybrid rice reported in previous studies, they failed to discuss their data sufficiently from this perspective. The authors attributed differences in plant performance between the two locations to soil fertility, yet provided no detailed soil fertility analysis to support this claim. Additionally, in the Abstract and Conclusion sections, the authors suggested they identified physiological characteristics that could break the yield ceiling in hybrid rice varieties. However, their data interpretation does not fully support these arguments. The authors should restructure their manuscript by incorporating clear, testable hypotheses.
Response 1: Agree. We have been revised the manuscript according to the reviewer’s suggestion, and the add the hypotheses in Line 130-135. The abstract and conclusion have been revised. Line 620- 627.
Comment 2: The statistical approach is inappropriate for the experimental design. Given the nested structure of the data, the analysis should employ either nested ANOVA or generalized linear mixed model. The results should be presented with appropriate error terms and significance levels.
Response 2: In our manuscript, data analysis were carried out with the the Statistix 8 software package (Analytical Software, Tallahassee, Florida, USA). For field experiments, the statistical model of the analysis of variance (ANOVA) included location, plant density, variety, and the interactions between these factors. Clarifying the effects of location, plant density, variety, and the interactions between these factors on grain yield and yield attributes of super hybrid rice; and isolate the individual contributions of latitudes and altitudes to the variations in super high rice yields. For the pot experiments, the ANOVA model similarly included location, soil, and variety, as well as the two-factor interactions L×S, L×V, and S×V, and the three-factor interaction L×S×V. Clarifying the effects of location, soil, and variety and the interactions between these factors on grain yield and yield attributes of super hybrid rice; and isolate the individual contributions of soil fertility and climatic conditions to the variations in super high rice yields. Correlation plot analysis was used to evaluate the relationships between grain yield, yield attributes and climatic factors (Origin 2024, OriginLab Corp., Northampton, MA, USA). Based on the results from both field and pot experiments, it is easily to isolate the individual contributions of altitude, latitude and soil fertility of planted site to the variations in super high rice yields based on the findings of both the paddy field and pot experiments. So, the statistical approach is appropriate.
Comment 3: All figures and tables require revision to be standalone. Current issues include 1) undefined abbreviations, 2) incomplete legends, and 3) absence of marginal means.
Response 3: We have been revised the manuscript according to the reviewer’s suggestion. Table 1, 2, 3, 4, 5 and 6.
Comment 4: The authors' emphasis on soil fertility requires more comprehensive analysis. The authors should 1) present soil characteristics in a detailed table, 2) describe soil sampling methodology and timing clearly, 3) document temporal changes in soil parameters, and 4) analyze relationships between soil properties and yield components.
Response 4: The soil characteristics were showed in table 9. Soil testing was performed on samples taken from the upper 20 cm layer before transplanting the rice seedlings in 2020. In this study, Hanyuan exhibited higher organic matter, total nitrogen, total phosphorus, total potassium, available nitrogen, available potassium compared to Luxian. Line 392-394, 409-413; Line 524-526.
Comment 5: L78-L82: Provide evidence to support these claims
Response 5: We have been deleted the sentences.
Comment 6: L97-L99: Explain the relationship between yield components and environmental conditions
Response 6: Increasing plant density is an effective approach to increasing panicles per unit land area that achieving high grain yields in southern China. In the present study, the yield components (panilce number, spikelets per pabnicle, spikelets per m2 and grain filling), pre- and post-heading biomass production, pre- and post-heading CGR and RUE were significantly negatively correlated with temperatures (except for the mean maximum temperature from transplanting to heading) but significantly positively correlated with cumulative solar radiation pre- and post-heading.
Comment 7: L119-L120: Include quantitative analysis of published data inconsistencies
Response 7: We have been revised the manuscript according to the reviewer’s suggestion. Line111-115.
Comment 8: L521-L530: Present soil characteristics in a comprehensive table
Response 8: The soil characteristics of the paddy field in two location were showed in Table 9. Page 17
Comment 9: L576-L577: Specify soil sampling timeline and frequency
Response 9: Soil testing was performed on samples taken from the upper 20 cm layer before transplanting the rice seedlings in 2020. Line 525-526
Comment 10: L193-L195: Revise for clarity and accuracy
Response 10: We have been revised the manuscript according to the reviewer’s suggestion. Line187-188
Comment 11: L365-L366: Verify yield comparison data
Response 11: We have checked the comparison data, the highest grain yield were 15.2, 14.1, 16.5 and 15.8 t ha-1 in the reference 8-10, 13, respectively. The highest grain yield in the present study is higher than those reported in earlier studies at high-altitude sites. Line370-371
Comment 12: L372-L373: Provide evidence for this statement
Response 12: The climatic conditions were showed in Table 1. We have been revised the manuscript according to the reviewer’s suggestion. Line 374-382.
Comment 13: L615-L616: Support claim with experimental data
Response 13:We have been revised the manuscript according to the reviewer’s suggestion. Line 615-627
Comment 14: The Introduction and Discussion sections contain redundant and irrelevant content that should be removed.
Response 14: We have been revised the manuscript according to the reviewer’s suggestion.The Introduction: Line79-80, 98-102, 111-115, 131-135; Th Discussion: Line 364-372, 374-384, 386-388, 392-394, 395-399, 409-413,462-466, 510-517.
Comment 15: The Abstract requires revision to eliminate ambiguous statements such as "provide valuable insights for enhancing super hybrid rice productivity" and "superior light conditions and temperatures compared to Luxian." The revised Abstract should clearly present the specific findings and conclusions supported by the data.
Response 15: We have been revised the manuscript according to the reviewer’s suggestion. Line 29-33, 38-39, 41-43
The revised manuscript has been submitted to your journal. We look forward to your positive responses. Thank you very much and best regards.
Sincerely yours,
Peng jiang
Rice and Sorghum Research Institute, Sichuan Academy of Agricultural Sciences
Deyang 618000, China
FuXian Xu
Rice and Sorghum Research Institute, Sichuan Academy of Agricultural Sciences
Deyang 618000, China

Reviewer 4 Report
Comments and Suggestions for Authors
Dear authors,
I understand that the authors revised the manuscript following the comments by referees. However, I think that there are more revisions required in the revised manuscript as attached.

Author Response
Dear reviewer and editors,
We thank for the reviewer’s valuable comments on the manuscript entitled “Yield performance of super hybrid rice grown on subtropical environments at similar latitude but different altitudes” (ID:plants-3304475). All comments are highly valuable for improving the manuscript. We have completely revised the manuscript in accordance with the reviewers’ comments. All changes are marked in red in the manuscript, and our point-by-point responses are listed below.
Comments 1: Abstract: Hanyuan (High-altitude at 1,000 m above sea level, asl.) and Luxian (low-altitude, 300 m asl.) should beÄ´ er be added for the reader to easily understand the difference in altitude between two sites.
Response 1: We have been revised the manuscript according to the revivewer’ suggestion. Line 21-22, 26, 29, 34 and 36.
Comments 2: I advised that the authors should beÄ´ er incorporate the longer growth duration linked with higher cumulative solar radiation before and after heading in Hanyuan into Lines 372–376, as follows:
Response 2: We have been revised the manuscript according to the revivewer’ suggestion. Line 378-382
The revised manuscript has been submitted to your journal. We look forward to your positive responses. Thank you very much and best regards.
Sincerely yours,
Peng jiang
Rice and Sorghum Research Institute, Sichuan Academy of Agricultural Sciences
Deyang 618000, China
FuXian Xu
Rice and Sorghum Research Institute, Sichuan Academy of Agricultural Sciences
Deyang 618000, China

Round 3
Reviewer 3 Report
Comments and Suggestions for Authors
None
Author Response
There are no other Comments and Suggestions in the Round 3 from the reviwer 3.